# The Distribution and Prevalence of High-Risk HPV Genotypes Other than HPV-16 and HPV-18 among Women Attending Gynecologists’ Offices in Kazakhstan

**DOI:** 10.3390/biology10080794

**Published:** 2021-08-18

**Authors:** Gulzhanat Aimagambetova, Aisha Babi, Alpamys Issanov, Sholpan Akhanova, Natalya Udalova, Svetlana Koktova, Askhat Balykov, Zhanna Sattarkyzy, Zhuldyz Abakasheva, Azliyati Azizan, Chee Kai Chan, Torgyn Issa

**Affiliations:** 1Department of Biomedical Sciences, School of Medicine, Nazarbayev University, Nur-Sultan 010000, Kazakhstan; aisha.mukushova@nu.edu.kz (A.B.); aazizan@touro.edu (A.A.); cchan@kean.edu (C.K.C.); torgyn.issa@nu.edu.kz (T.I.); 2Department of Medicine, School of Medicine, Nazarbayev University, Nur-Sultan 010000, Kazakhstan; alpamys.issanov@nu.edu.kz; 3Obstetrics and Gynecology Department, Keruen Medicus Clinic, Almaty 050040, Kazakhstan; info@keruen-medicus.kz; 4Obstetrics and Gynecology Department, East Kazakhstan Regional Hospital, Oskemen 07000, Kazakhstan; udalovanm@mail.ru; 5Regional Perinatal Center, Altynsarin 3, Aktobe 030006, Kazakhstan; svetlana.koktova@gmail.com (S.K.); balykov.askhat@gmail.com (A.B.); 6Obstetrics and Gynecology Department, City polyclinic #6, Nur-Sultan 010000, Kazakhstan; dr.sattarkyzy@list.ru; 7Gynecology Department, Daliya Clinic, Pavlodar 140013, Kazakhstan; zhyldiz_76@mail.ru; 8Department of Basic Sciences, College of Osteopathic Medicine, Touro University Nevada, Henderson, NV 89014, USA; 9Department of Biology, College of Science and Technology, Wenzhou-Kean University, Wenzhou 325000, China

**Keywords:** cervical cancer, HPV, high-risk HPV, HPV epidemiology, Kazakhstan

## Abstract

**Simple Summary:**

This study focused on the prevalence of high-risk human papilloma virus (HR-HPV) infection types, other than HPV-16 and -18, in women throughout Kazakhstan due to the high rates of cervical cancer in Kazakhstani women. Approximately one quarter of the of the participants were infected with HR-HPV types other than HPV-16 and -18; 72% of these women were infected with one HR-HPV type with HPV-31 being the most prevalent, and the remaining 28% of these infected women were infected with multiple HR-HPVs with HPV-68 being the most prevalent type in these women. Introducing HR-HPV testing for all 14 cancerogenic types into cervical cancer screening program could help to reduce the rates of cervical cancer in Kazakhstan.

**Abstract:**

Cervical cancer represents a great burden to public health of women. This study aimed to obtain a nationwide genotyping survey and analysis of high risk-HPV including those that are caused by HPV types other than HPV-16 and HPV-18, among women in Kazakhstan. This study was conducted based on the collection of survey and cervical swabs of 1645 women across the country. The samples were genotyped for high-risk HPV types based on real-time PCR methods. Collected data was analyzed with the focus on high-risk HPV types other than HPV-16 and -18. Infection was present in 22% of women who participated in the study. The most prevalent types were HPV-31 among single infections and HPV-68 among multiple infections. Conclusively, despite the lack of attention high-risk HPV types beyond HPV-16 and -18 get in attempts of cervical cancer prevention in Kazakhstan, their prevalence is high and plays a large role in cervical cancer epidemiological situation.

## 1. Introduction

Cervical cancer is the leading cause of cancer-related deaths in developing countries. Moreover, cervical cancer is estimated to be the fourth most common cancer in women worldwide [1]. Majority of cervical cancer cases are related to human papillomavirus (HPV) infection. High-risk (HR) HPV types -16 and -18 are responsible for about 70% of all cervical cancer cases worldwide. As a result, diagnosis and management of HPV-16 and HPV-18 dominate medical research [2]. However, other HR-HPV genotypes also play a significant role in cervical cancer epidemiology and are responsible for about 20% of cervical cancer cases [3].

Despite the presence of the national cervical cancer screening program in Kazakhstan, the disease incidence has increased during the recent decade. The annual rate of cervical cancer incidence in the country increased from 16.3 ± 0.4 per 100,000 female population in 2009 to 19.5 ± 0.5 in 2018 [4]. This trend could be linked to several factors, such as low coverage of the cervical cancer screening program (around 46%) and the absence of HR-HPV screening. Moreover, gynecologists in Kazakhstan mainly give patient referrals to test for HPV-16 and HPV-18, causing the majority of Kazakhstani clinical laboratories to intensely focus on these two HR-HPV types [5]. Knowledge on the distribution of all HR-HPV types could help patients choose proper vaccine strategies and push forward efforts to implement HPV primary infection prevention protocols [2,6]. 

Previous HR-HPV genotyping studies conducted in Kazakhstan include both HPV-16 and HPV-18 and HR-HPV types other than 16 and 18 [5,6]. However, there is still limited data available on the real burden of HPV infection in Kazakhstan [2,6]. Therefore, in an effort to obtain more accurate information regarding HPV prevalence and related practices, we conducted this study to initiate a nationwide genotyping analysis of HR-HPV, including types other than HPV-16 and -18, among Kazakhstani women.

## 2. Materials and Methods

A prospective cross-sectional study among Kazakhstani women was conducted from May 2019 until December 2020. Sample collection was conducted in five major cities from different regions of Kazakhstan (South, North, East, West and the Capital City regions).

After obtaining informed consent, only patients who agreed to participate were included in the study. The following inclusion criteria were used to choose women for study participation: (1) age from 18 to 70; (2) attends a gynecological center located in one of chosen cities for the study; and (3) able to fill questionnaire in Kazakh, Russian or English language. Each woman participating in the study had a Papanicolaou (Pap) smear test performed as gynecologists were taking cervical swabs. Gynecologists also recorded the age of the patients along with other demographic data (Appendix A are available online).

The samples were collected into 1.5 mL Eppendorf tubes and were kept frozen until further use. Wizard^®^ Genomic DNA Purification Kit was used for DNA extraction following the manufacturer’s manual. Genotyping was carried out by PCR using the AmpliSens^®^ HPV HCR genotype-titer-FRT kit (InterLabService, Moscow, Russia) that identifies 14 high-risk HPV types (16, 18, 31, 33, 35, 39, 45, 51, 52, 56, 58, 59, 66, 68) on the CFX 96 Real-Time PCR machine (Bio-Rad Laboratories). Genotyping was performed following the instructions of the manufacturer, including positive and negative controls, samples concentration and thresholds for positivity of the samples. 

The data collected was analyzed using STATA 16 software [7]. Prevalence of HPV genotypes, descriptive statistics and one way analysis of variances were calculated. 

The study was approved by the Institutional Research Ethics Committee of Nazarbayev University (NU IREC) on 23 April 2019 (IREC number: 146/4042019). All study participants were given information on the risks, benefits, goals and methods of the study. Confidentiality of the collected data was protected.

## 3. Results

In total 1645 women participated in the study. The mean age of the participants was 35.98 ± 9.97, and 40.49% of women were aged between 26 and 35. The minimum age was 18 years and the maximum age was 70 years. More than three quarters (76.25%) of the participants identified themselves as ethnic Kazakh. There was almost equal regional distribution of women who participated in the study (around 20% from each region of the country) with the exception of the north region (around 16%). Almost half (47.17%) of the participants had graduate-level education and more than one quarter (34.77%) of women had an undergraduate-level diploma. A majority of women (82.07%) were married or in a committed relationship. More than three quarters of the participating women (79.45%) had one or more children 

Out of the total number of participants, 22.49% (*n* = 370) were infected with HR-HPV other than HPV-16 or 18, which amounts to 71.62% (*n* = 265) of HPV positive women or 16% of all study participants. These women had a single infection with only one HR-HPV type. Among HR-HPV types, excluding HPV-16 and 18, HPV-31 (18%), HPV-51 (14%), HPV-68 (11%) and HPV-52 (9%) were the most prevalent single HR-HPV infection (Table 1). Infection with multiple HR-HPV genotypes other than HPV-16 and -18 was seen in 105 (28%) of HPV positive women or 6% of all the study participants. Among participants with multiple HR-HPV infections, HPV-68 was present in 33%, HPV-39 in 31%, HPV-52 in 31%, HPV-35 in 25% and HPV-51 in 24% of the sample population (Table 1). 

In terms of age distribution, the HR-HPV genotypes were most frequently detected in the 26–35 age group, followed by 36–45 age group (Table 1). In both single and multiple HR-HPV infection groups, the lowest prevalence was in the youngest and in the oldest age groups, with the exception of HPV-51, HPV-58, HPV-59 and HPV-66 where the highest prevalence was either in the youngest or in the oldest age group.

The geographical distribution of HR-HPV types other than HPV-16 and HPV-18 is presented in Figure 1.

## 4. Discussion

To our knowledge, this is the first study that focuses on the distribution and prevalence of HR-HPV genotypes other than HPV-16 and HPV-18 in the different regions of Kazakhstan. Moreover, this study looked at the prevalence of single and multiple HR-HPV infections among the participants. 

The results of this investigation showed that 22% of women included in the study were positive for one or more HR-HPV types other than HPV-16 and HPV-18, whereby 16% had a single HR-HPV infection and 6% had multiple HR-HPV infection. When comparing to the results of previous study [8]—where HPV-16 had highest prevalence in both single and multiple infections (54% and 57%, respectively), and HPV-18 had prevalence of 6% and 13%—the distribution of other HR-HPV genotypes was more uniform. Only HPV-68 was highly prevalent in both single and multiple HR-HPV infections. HPV-59, HPV-31 and HPV-58 were mostly prevalent in single HR-HPV infection, while HPV-39, HPV-52, HPV-35 and HPV-51 were mostly prevalent in multiple HR-HPV infections. 

We found it difficult to compare HR-HPV prevalence with previous studies conducted in Kazakhstan, as these studies have included HPV-16 and HPV-18 [5,6] and presented data was limited to some Kazakhstani regions and did not show the nationwide prevalence. Another challenge to perform a comparative analysis of the findings is that previous studies did not specify multiple and single infections while reporting the results. The study conducted in 2016 in the western region of the country found 26% HR-HPV positive cases [5]. The most prevalent HR-HPV genotypes other than HPV-16 and HPV-18 found in that study, were HPV-39 (5.8%), HPV-51 (5.3%) and HPV-31 (4.9%) [5]. In our pilot study conducted in 2017 the overall HR-HPV prevalence was 43.6% and HPV-33, HPV-51 and HPV-52 were the most prevalent [6]. Another study published in 2019 found 25% HR-HPV positive patients, and among the most prevalent HR-HPV genotypes were HPV-33 (4.9%), HPV-51 (4.9%) and HPV-52 (4.9%) [9]. 

There are both similarities and differences in the distribution of HR-HPV genotypes other than HPV-16 and HPV-18 in the studies conducted in Kazakhstan: HPV-31, HPV-39, HPV-51 and HPV-52 were found to be the most prevalent. However, unlike previous reports, this study found a high prevalence of HPV-35, HPV-59, HPV-58 and HPV-68. Therefore, even within one country HR-HPV genotype distribution can vary significantly among regions. 

Our study also found that women at the reproductive age had the highest prevalence of HR-HPV infection. Women aged between 26 and 35 had the highest HR-HPV prevalence, followed by the age group between 36–45 years. This finding is similar to the previous studies, which found that HPV prevalence was highest in the young age, and HPV prevalence was lower among women aged older than 60 [10,11]. In contrast, one study found that older women were more likely to have high HPV prevalence [12].

The strengths of this study include large sample size, wide-geographical distribution of collected data including western, northern, eastern and southern regions, and differentiation between single and multiple HR-HPV infection, as well as a focus on detecting HR-HPV genotypes other than HPV-16 and HPV-18. However, the study results should be interpreted and generalized cautiously because the participants were non-randomly selected, using the convenience sampling technique.

Currently, the cervical cancer prevention program in Kazakhstan does not include screening for HPV infection or HPV vaccination as a primary prevention. Moreover, as HPV testing is not covered by the government or insurance, and investigation of only HPV-16 and 18 is a cheaper option, many gynecologists do not require their patients to test for other HR-HPV types with the financial burden in mind. More research needs to be conducted to study prevalence of all HR-HPV types including different cervical intra-epithelial neoplasia stages. Nationwide HPV genotyping among the general population in Kazakhstan including both male and female patients is required to have a full picture of HR-HPV burden. Additionally, the national cervical cancer prevention program should be expanded beyond regular Pap smear tests.

## 5. Conclusions

To conclude, this study focused on genotyping analysis of HR-HPV other than HPV-16 and HPV-18 among women attending gynecologists’ offices in Kazakhstan. Despite the fact that other HR-HPV genotypes are responsible for less numbers of cervical cancer, they still play an important role in the epidemiology of the disease. Introducing compulsory testing for HR-HPV types other than HPV-16 and HPV-18 and more attention on the problem from gynecology and oncology specialists could help to reduce the increasing rates of cervical cancer in Kazakhstan.

## Figures and Tables

**Figure 1 biology-10-00794-f001:**
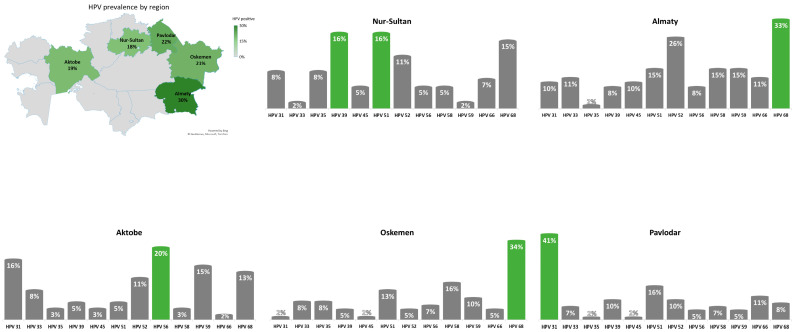
Geographical distribution of HR-HPV types other than HPV-16 and HPV-18 across Kazakhstan.

**Table 1 biology-10-00794-t001:** High-risk HPV genotypes distribution among single and multiple high-risk HPV infection by age groups (*n* = 370).

HR-HPV * Type	Single HR-HPV Infection, %				Total N (%) 265 (100%)	Multiple HR-HPV Infection, %				Total N (% ^1^) 105 (100%)
	18–25 Years	26–35 Years	36–45 Years	45+ Years		18–25 Years	26–35 Years	36–45 Years	45+ Years	
HPV-31	2%	49%	28%	21%	47 (18%)	26%	52%	22%	0	23 (22%)
HPV-33	10%	45%	40%	5%	20 (8%)	33%	50%	17%	0	12 (11%)
HPV-35	0	64%	27%	9%	11 (4%)	31%	31%	27%	11%	26 (25%)
HPV-39	18%	32%	36%	14%	22 (8%)	30%	37%	27%	6%	33 (31%)
HPV-45	20%	40%	30%	10%	10 (4%)	8%	50%	0	12%	8 (8%)
HPV-51	24%	43%	22%	11%	37 (14%)	32%	32%	32%	4%	25 (24%)
HPV-52	24%	60%	12%	4%	25 (9%)	24%	52%	15%	9%	33 (31%)
HPV-56	6%	50%	33%	11%	18 (7%)	25%	33%	42%	0	12 (11%)
HPV-58	21%	42%	21%	16%	19 (7%)	44%	31%	13%	12%	16 (15%)
HPV-59	0	29%	42%	29%	17 (6%)	11%	63%	26%	0	19 (18%)
HPV-66	9%	46%	36%	9%	11 (4%)	47%	33%	20%	0	15 (14%)
HPV-68	4%	25%	54%	17%	28 (11%)	20%	37%	37%	6%	35 (33%)

^1^ percentage of HR-HPV type in multiple infection represents prevalence of the type in participants with multiple infection. * HR-HPV—high-risk human papillomavirus

## Data Availability

The study questionnaires and raw data are available via the link: https://zenodo.org/record/4600664#.YEsv6WgzbIU (accesses on 21 April 2021).

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
