# Peer review of "The Distribution and Prevalence of High-Risk HPV Genotypes Other than HPV-16 and HPV-18 among Women Attending Gynecologists’ Offices in Kazakhstan"

_biology, 2021, doi:10.3390/biology10080794_

Round 1
Reviewer 1 Report
Aimagambetova conducted a study focused on genotyping analysis of high-risk HPV other than HPV-16 and HPV-18 from cervical swabs of 1645 women in Kazakhstan. They found that infection was present in 22% of women and the most prevalent types were HPV-31 among single infections and HPV-68 among multiple infections. This is an interesting work and could help to reduce the increasing rates of cervical cancer in Kazakhstan.
Only one minor comment:
For the clinical specimens, it would be nice to include some clinical information such as age, clinical stage, etc.
Author Response
Dear Reviewer,
Thank you very much for the detailed review of our manuscript. We appreciate a lot your time, efforts, valuable comment, and suggestions that helped us to improve the quality of the text. Please find below our response to all your comment.
Comments and Suggestions for Authors
Aimagambetova conducted a study focused on genotyping analysis of high-risk HPV other than HPV-16 and HPV-18 from cervical swabs of 1645 women in Kazakhstan. They found that infection was present in 22% of women and the most prevalent types were HPV-31 among single infections and HPV-68 among multiple infections. This is an interesting work and could help to reduce the increasing rates of cervical cancer in Kazakhstan.
Only one minor comment:
For the clinical specimens, it would be nice to include some clinical information such as age, clinical stage, etc.
Response – thank you for the comment.
The following available patients’ socio-demographic data have been reported. There were no women with cervical cancer recruited to the study, therefore, we cannot present clinical staging.
“In total 1645 women participated in the study. The mean age of the participants was 35.98±9.97, and 40.49% of women were aged between 26 and 35. The minimum age was 18 years and the maximum age was 70 years. More than three quarters (76.25%) of the participants identified themselves as ethnic Kazakh. There was almost equal regional distribution of women participated in the study (around 20% from each region of the country) with the exception in north region (around 16%). Almost half (47.17%) of the participants had graduated from the university and more than one quarter (34.77%) of women had received technical or vocational diploma. Majority of women (82.07%) were married or in a committed relationship. More than three quarters of participating women (79.45%) had one or more children “.

Reviewer 2 Report
Overall, I think there is enough data and interest here to warrant a Brief Report. However, I think the paper needs extensive editing of English language and style, especially to make clear what the purpose of this study is to the readers. The use of apostrophes is not recommended, and "high risk HPV" should be abbreviated to "HR-HPV" to help condense the text.
Here's my major English correction requests:
Lines 27-32:
This study focused on the prevalence of high-risk human papilloma virus (HR-HPV) infection types, other than HPV-16 and -18, in women throughout Kazakhstan due to the high rates of cervical cancer in Kazakhstani women. Approximately one quarter of the of the participants were infected with HR-HPV types other than HPV-16 and -18; 72% of these women were infected with one HP-HPV type, with HPV-31 being the most prevalent, and the remaining 28% of these infected women were infected with multiple HR-HPVs, with HPV-68 being the most prevalent type in these women. Introducing HR-HPV testing for all 14 cancerogenic types into cervical cancer screening program could help to reduce the rates of cervical cancer in Kazakhstan.
Lines 34-35:
Cervical cancer represents a great burden to public health of women.
Lines 50-52:
Majority of cervical cancer cases are related to human papillomavirus (HPV) infection. High-risk (HR) HPV types -16 and -18 are responsible for about 70% of all cervical cancer cases worldwide. As a result, diagnosis and management of HPV-16 and HPV-18 dominate medical research [2]
Line 53:
However, other HR-HPV genotypes also play a significant…
Line 60-61:
and the absence of HR-HPV screening….
Lines 62-80 (rewrite, compact some text):
Moreover, gynecologists in Kazakhstan mainly give patient referrals to test for HPV-16 and HPV-18, causing the majority of Kazakhstani clinical laboratories to intensely focus on these two HR-HPV types. Knowledge on the distribution of all HR-HPV types could help patients choose proper vaccine strategies and push forward efforts to implement HPV primary infection prevention protocols
There is still limited data available on the real burden of HPV infection in Kazakhstan. Therefore, in an effort to obtain more accurate information regarding HPV prevalence and related practices, we conducted this study to initiate a nationwide genotyping analysis of HR-HPV, including types other than HPV-16 and -18, among Kazakhstani women.
Lines 82-95 (rewrite):
A prospective cross-sectional study among Kazakhstani women was conducted from May 2019 until December 2020. Sample collection was conducted in five major cities from different regions of Kazakhstan (South, North, East, West and the Capital City regions).
After obtaining an informed consent, only patients who agreed to participate were included in the study. The following inclusion criteria were used to choose women for study participation: (1) age from 18 to 70; (2) attends a gynecological center located in one of chosen cities for the study; (3) able to fill questionnaire on Kazakh, Russian or English language. Each woman participating in the study had a Papanicolaou (Pap) smear test performed as gynecologists were taking cervical swabs. Gynecologists also recorded the age of the patients along with other demographic data.
Genotyping was performed following the instructions of the manufacturer….
Lines 108-113
In total, 1,645 women in Kazakhstan participated in this study. Out of the total number of participants, 22% (n=370) were infected with HR-HPV, other than HPV-16 or 18, which amounts to 72% (n=265) of HPV positive women or 16% of all study participants. These women had single infection with only one HR-HPV type. Among HR-HPV types, excluding HPV-16 and 18, HPV-31 (18%), HPV-51 (14%), HPV-68 (11%), and HPV-52 (9%) were the most prevalent single high-risk HPV infection (Table 1)
There is one more addition that could help increase the interest of this paper - a second table or figure that shows the geological distribution of the different HR-HPVs. Is the overall amount of different HR-HPVs the same in all women tested across all Kazakhstan region or do some HR-HPV infections predominate in some regions and not in others? Since the data has already been collected, it should be relatively straight-forward to make such a figure.
Author Response
Dear Reviewer,
Thank you very much for the detailed review of our manuscript. We appreciate a lot your time, efforts, valuable comments, and suggestions that helped us to improve the quality of the text. Please find below our point-by-point responses to all your comments.
Comments and Suggestions for Authors
Overall, I think there is enough data and interest here to warrant a Brief Report. However, I think the paper needs extensive editing of English language and style, especially to make clear what the purpose of this study is to the readers. The use of apostrophes is not recommended, and "high risk HPV" should be abbreviated to "HR-HPV" to help condense the text.
Here's my major English correction requests:
Lines 27-32:
This study focused on the prevalence of high-risk human papilloma virus (HR-HPV) infection types, other than HPV-16 and -18, in women throughout Kazakhstan due to the high rates of cervical cancer in Kazakhstani women. Approximately one quarter of the of the participants were infected with HR-HPV types other than HPV-16 and -18; 72% of these women were infected with one HP-HPV type, with HPV-31 being the most prevalent, and the remaining 28% of these infected women were infected with multiple HR-HPVs, with HPV-68 being the most prevalent type in these women. Introducing HR-HPV testing for all 14 cancerogenic types into cervical cancer screening program could help to reduce the rates of cervical cancer in Kazakhstan.
Response: Thank you for the comment. The text has been rewritten according to the suggestions.
“This study focused on the prevalence of HR-HPV types other than HPV-16 and HPV-18 types in different regions of Kazakhstan given the fact of high rates of cervical cancer in Kazakhstan. The study was conducted among women attending gynecologists’ offices. Less than one quarter of the participants had HR-HPV infection other than HPV-16 and 18. Among single HR-HPV infection the most prevalent was HPV-31. Among multiple HR-HPV infections the most prevalent was HPV-68. Introducing HR-HPV testing for all 14 cancerogenic types into cervical cancer screening program could help to reduce the rates of cervical cancer in Kazakhstan.”
New version: “This study focused on the prevalence of high-risk human papilloma virus (HR-HPV) infection types, other than HPV-16 and -18, in women throughout Kazakhstan due to the high rates of cervical cancer in Kazakhstani women. Approximately one quarter of the of the participants were infected with HR-HPV types other than HPV-16 and -18; 72% of these women were infected with one HP-HPV type, with HPV-31 being the most prevalent, and the remaining 28% of these infected women were infected with multiple HR-HPVs, with HPV-68 being the most prevalent type in these women. Introducing HR-HPV testing for all 14 cancerogenic types into cervical cancer screening program could help to reduce the rates of cervical cancer in Kazakhstan.”
Lines 34-35:
Cervical cancer represents a great burden to public health of women.
Response: Thank you for the comment. The sentence is corrected.
“Cervical cancer represents a great burden to public health, particularly affecting health of women.”
Now: “Cervical cancer represents a great burden to public health of women.”
Lines 50-52:
Majority of cervical cancer cases are related to human papillomavirus (HPV) infection. High-risk (HR) HPV types -16 and -18 are responsible for about 70% of all cervical cancer cases worldwide. As a result, diagnosis and management of HPV-16 and HPV-18 dominate medical research [2]
Response: Thank you for the suggestion. The text “Majority of cervical cancer cases are related to human papillomavirus infection (HPV). High-risk HPV types 16 and 18 are responsible for about 70% of all cervical cancer cases worldwide. As a result diagnosis and management of HPV-16 and HPV-18 dominate medical research [2]” is corrected and now sounds as follows “Majority of cervical cancer cases are related to human papillomavirus (HPV) infection. High-risk (HR) HPV types -16 and -18 are responsible for about 70% of all cervical cancer cases worldwide. As a result, diagnosis and management of HPV-16 and HPV-18 dominate medical research [2].”
Line 53:
However, other HR-HPV genotypes also play a significant…
Response: Thank you for the comment. Corrected.
Line 60-61:
and the absence of HR-HPV screening….
Response: The sentence is corrected.
Lines 62-80 (rewrite, compact some text):
Moreover, gynecologists in Kazakhstan mainly give patient referrals to test for HPV-16 and HPV-18, causing the majority of Kazakhstani clinical laboratories to intensely focus on these two HR-HPV types. Knowledge on the distribution of all HR-HPV types could help patients choose proper vaccine strategies and push forward efforts to implement HPV primary infection prevention protocols
There is still limited data available on the real burden of HPV infection in Kazakhstan. Therefore, in an effort to obtain more accurate information regarding HPV prevalence and related practices, we conducted this study to initiate a nationwide genotyping analysis of HR-HPV, including types other than HPV-16 and -18, among Kazakhstani women.
Response: Thank you for the comment. The text in lines 62-80 was rewritten as suggested by Reviewer.
“Moreover, in Kazakhstan gynecologists give a referral for the patients mainly to test for HPV-16 and HPV-18. As a result, the majority of the clinical laboratories in Kazakhstan perform HPV genotyping test mainly on HPV-16 and HPV-18 [5].
Although current worldwide trends carefully explore all high-risk HPV types including other than 16 and 18, there were no published studies conducted in Kazakhstan focusing on the types distribution of HPV infection other than 16 and 18 [5]. Moreover, there is no national HPV vaccination program in Kazakhstan. As prevalence of high-risk HPV types other than 16 and 18 are significant in diagnostics, management, and vaccination strategies [7], it is important to consider research with a focus on all high-risk HPV types. Knowledge on all high-risk types’ distribution could help to choose proper vaccine and push forward the efforts on the implementation of HPV infection primary prevention.
Despite presence of some studies on high-risk HPV topic in Kazakhstan, still limited data are available on the real burden of HPV infection. Therefore, in an effort to obtain more accurate information regarding HPV prevalence and related practices, the aim of this study was to conduct a nationwide genotyping analysis of high-risk HPV including those that that are not HPV-16 and HPV-18, among women attending gynecologists’ offices in Kazakhstan.“
Now it is as follows:
‘Moreover, gynecologists in Kazakhstan mainly give patient referrals to test for HPV-16 and HPV-18, causing the majority of Kazakhstani clinical laboratories to intensely focus on these two HR-HPV types [5]. Knowledge on the distribution of all HR-HPV types could help patients choose proper vaccine strategies and push forward efforts to implement HPV primary infection prevention protocols [6, 7].
Previous HR-HPV genotyping studies conducted in Kazakhstan include both HPV-16 and HPV-18 and HR-HPV types other than 16 and 18 [5,6]. However, there is still limited data available on the real burden of HPV infection in Kazakhstan [6,7]. Therefore, in an effort to obtain more accurate information regarding HPV prevalence and related practices, we conducted this study to initiate a nationwide genotyping analysis of HR-HPV, including types other than HPV-16 and -18, among Kazakhstani women.”
Lines 82-95 (rewrite):
A prospective cross-sectional study among Kazakhstani women was conducted from May 2019 until December 2020. Sample collection was conducted in five major cities from different regions of Kazakhstan (South, North, East, West and the Capital City regions).
After obtaining an informed consent, only patients who agreed to participate were included in the study. The following inclusion criteria were used to choose women for study participation: (1) age from 18 to 70; (2) attends a gynecological center located in one of chosen cities for the study; (3) able to fill questionnaire on Kazakh, Russian or English language. Each woman participating in the study had a Papanicolaou (Pap) smear test performed as gynecologists were taking cervical swabs. Gynecologists also recorded the age of the patients along with other demographic data.
Genotyping was performed following the instructions of the manufacturer….
Response: Thank you for the comment. The text in lines 62-80 was rewritten as suggested by Reviewer.
The following text “A prospective cross-sectional study among women attending gynecologists’ offices was conducted from May 2019 until December 2020. Sample collection was conducted in five major cities from different regions of Kazakhstan (south, north, east, west and the capital city regions).
After obtaining an informed consent, only patients who agreed to participate were included in the study. Eligible to participate in the study women were those who met the following inclusion criteria: (1) age from 18 to 70; (2) gynecological offices attenders in one out of chosen cities for the study; (3) able to fill questionnaire on Kazakh, Russian or English language. Each women participating in the study had Papanicolaou (Pap) smear testing as gynecologists were taking cervical swabs. Gynecologists also recorded patients’ age and other demographic data. “
was rewritten, and now it will be presented as follows:
“A prospective cross-sectional study among Kazakhstani women was conducted from May 2019 until December 2020. Sample collection was conducted in five major cities from different regions of Kazakhstan (South, North, East, West and the Capital City regions).
After obtaining an informed consent, only patients who agreed to participate were included in the study. The following inclusion criteria were used to choose women for study participation: (1) age from 18 to 70; (2) attends a gynecological center located in one of chosen cities for the study; (3) able to fill questionnaire on Kazakh, Russian or English language. Each woman participating in the study had a Papanicolaou (Pap) smear test performed as gynecologists were taking cervical swabs. Gynecologists also recorded the age of the patients along with other demographic data.”
Lines 108-113
In total, 1,645 women in Kazakhstan participated in this study. Out of the total number of participants, 22% (n=370) were infected with HR-HPV, other than HPV-16 or 18, which amounts to 72% (n=265) of HPV positive women or 16% of all study participants. These women had single infection with only one HR-HPV type. Among HR-HPV types, excluding HPV-16 and 18, HPV-31 (18%), HPV-51 (14%), HPV-68 (11%), and HPV-52 (9%) were the most prevalent single high-risk HPV infection (Table 1)
Response: The text in the lines 108-113 was rewritten according to the suggestions.
The original text “In total 1,645 women attending gynecologist’s offices participated in the study. Out of the total number of participants, 22% (n=370) were infected with high-risk HPV other than HPV-16 or 18, which amounts to 72% (n=265) of HPV positive women or 16% of the all study participants, who had single infection with only one HPV type. Among high-risk HPV types, excluding HPV-16 and 18, HPV-31 (18%), HPV-51 (14%), HPV-68 (11%), and HPV-52 (9%) were the most prevalent single high-risk HPV infection (Table 1)
was recomposed and now present according to the reviewer’s suggestions as follows
“In total, 1,645 women in Kazakhstan participated in this study. Out of the total number of participants, 22% (n=370) were infected with HR-HPV, other than HPV-16 or 18, which amounts to 72% (n=265) of HPV positive women or 16% of all study participants. These women had single infection with only one HR-HPV type. Among HR-HPV types, excluding HPV-16 and 18, HPV-31 (18%), HPV-51 (14%), HPV-68 (11%), and HPV-52 (9%) were the most prevalent single high-risk HPV infection (Table 1).”
There is one more addition that could help increase the interest of this paper - a second table or figure that shows the geological distribution of the different HR-HPVs. Is the overall amount of different HR-HPVs the same in all women tested across all Kazakhstan region or do some HR-HPV infections predominate in some regions and not in others? Since the data has already been collected, it should be relatively straight-forward to make such a figure.
Response: Thank you for the suggestion. The figure to show the geographical distribution has been created.

Reviewer 3 Report
The manuscript by Dr. Aimagambetova and colleagues addresses a very important question of HPV/Cervical cancer biology. Understanding and identifying what HPV genotypes are afflicting the community is crucial to proper management/vaccination.
As written, the manuscript is confusing to read. There needs to be a clearer statement as to what the percentiles mean. I found myself wondering if this was comparing to 16 or 18? and I couldn't tell what %age of the patients had an HPV infection, regardless of type. The conversation on the age of patients would benefit by stating how many of each age group (%ae of study). I think it would be easier to follow if the stats for HPV16 and 18 were included (for example, always state "in the full cohort X% were positive fo 16, Y% for 18, Z% for 31...etc. And were there any HPV16 and other genotypes found? or were HPV16 and HPV18 alone exclusively?
Author Response
Dear Reviewer,
Thank you very much for the detailed review of our manuscript. We appreciate a lot your time, efforts, valuable comments, and suggestions that helped us to improve the quality of the text. Please find below our responses to all your comments.
Comments and Suggestions for Authors
The manuscript by Dr. Aimagambetova and colleagues addresses a very important question of HPV/Cervical cancer biology. Understanding and identifying what HPV genotypes are afflicting the community is crucial to proper management/vaccination.
As written, the manuscript is confusing to read. There needs to be a clearer statement as to what the percentiles mean. I found myself wondering if this was comparing to 16 or 18? and I couldn't tell what %age of the patients had an HPV infection, regardless of type. The conversation on the age of patients would benefit by stating how many of each age group (%ae of study). I think it would be easier to follow if the stats for HPV16 and 18 were included (for example, always state "in the full cohort X% were positive fo 16, Y% for 18, Z% for 31...etc. And were there any HPV16 and other genotypes found? or were HPV16 and HPV18 alone exclusively?
Response: thank you for the comments. The manuscript has been rewritten to make it more clear and easy understandable to the potential readers. We have included available socio-demographic data (mean age, ethnicity, education, etc.) and hope that it made the manuscript more interesting. We have also created a figure to show the regional distribution of HR-HPV types across the country.
As the main goal of this paper is to report data on a nationwide genotyping analysis of HR- HPV types other that HPV-16 and HPV-18, we did not discuss a lot on HPV-16 and HPV-18. The prevalence of these two HR-HPV types have been reported in previous studies.
